# Factors associated with accelerometer measured movement behaviours among White British and South Asian children aged 6–8 years during school terms and school holidays

Liana Carmen Nagy,[1,2,3] Muhammad Faisal,[2,4] Maria Horne,[5] Paul Collings,[3,6] Sally Barber,[3] Mohammed Mohammed[2]

¹Oxford Brookes University, Faculty of Health and Life Sciences, Oxford, UK
²Faculty of Health Studies, University of Bradford, Bradford, UK
³Bradford Institute for Health Research, Bradford, UK
⁴Yorkshire & Humberside Academic Health Sciences Network, Wakefield, UK
⁵School of Healthcare, University of Leeds, Faculty of Medicine and Health, Leeds, UK
⁶Health Sciences, University of York, York, UK

**Correspondence to**
Liana Carmen Nagy;
lnagy@brookes.ac.uk

## ABSTRACT

**Objectives** To investigate factors associated with movement behaviours among White British (WB) and South Asian (SA) children aged 6–8 years during school terms and holidays.

**Design** Cross-sectional.

**Setting** Three primary schools from the Bradford area, UK.

**Participants** One hundred and sixty WB and SA children aged 6–8 years.

**Primary and secondary outcomes** Sedentary behaviour (SB), light physical activity (LPA) and moderate-to-vigorous physical activity (MVPA) measured by accelerometry during summer, winter and spring and during school terms and school holidays. Data were analysed using multivariate mixed-effects multilevel modelling with robust SEs. Factors of interest were ethnicity, holiday/term, sex, socioeconomic status (SES), weight status, weekend/weekday and season.

**Results** One hundred and eight children (67.5%) provided 1157 valid days of data. Fifty-nine per cent of children were WB (n=64) and 41% (n=44) were SA. Boys spent more time in MVPA (11 min/day, p=0.013) compared with girls and SA children spent more time in SB (39 min, p=0.017) compared with WB children in adjusted models. Children living in higher SES areas were more sedentary (43 min, p=0.006) than children living in low SES areas. Children were more active during summer (15 min MVPA, p<0.001; 27 LPA, p<0.001) and spring (15 min MVPA, p=0.005; 38 min LPA, p<0.001) and less sedentary (−42 min and −53 min, p<0.001) compared with winter. Less time (8 min, p=0.012) was spent in LPA during school terms compared with school holidays. Children spent more time in MVPA (5 min, p=0.036) during weekend compared with weekdays. Overweight and obese children spent more time in LPA (21 min, p=0.021) than normal-weight children.

**Conclusion** The results of our study suggest that significant child level factors associated with movement behaviours are ethnicity, sex, weight-status and area SES. Significant temporal factors are weekends, school holidays and seasonality. Interventions to support health enhancing movement behaviours may need to be tailored around these factors.

## Strengths and limitations of this study

► First study in Europe to evaluate accelerometer-measured physical activity in school terms and school holidays.
► Includes White British and South Asian children.
► Participants selected from high and low socioeconomic areas.
► The use of accelerometers underestimate water-based activities.
► The demographics of the population might limit the generalisability of the results.

## INTRODUCTION

High levels of sedentary behaviour (SB) and low levels of physical activity (PA) are common among primary school children and have become a public health concern due to negatively impacting physical and mental health.[1–3]

To date, studies have reported lower levels of PA and higher levels of SB in South Asian (SA) adults and children when compared with White British (WB) adults and children.[4–10] Often these differences have been explained on the account of cultural preferences, socioeconomic factors, lack of intrinsic motivation, lack of understanding about the benefits of PA and lack of understanding how to integrate PA in daily routine.[11 12] Within the UK, the majority of studies in the field of child PA and SB have been conducted with WB children[13–17] and very few have considered ethnic differences between WB and SA children.[9 18 19] The SA population accounts for ~8% of the UK population and the percentage rises up to 44% in various cities in the UK.[20]

More recently, researchers in the field of movement behaviour have started to

investigate different compositions of behaviours to identify those compositions which are most health enhancing.[21–24] For example, Saunders et al[25] investigated a reduction in light physical activity (LPA) or SB versus an increase in moderate-to-vigorous physical activity (MVPA) and concluded that children characterised by low SB, high PA and high levels of sleep have more desirable health outcomes.[25] Public health messages are consistent with these insights and have started adopting slogans such as 'Move more, sit less' to promote positive health behaviours in adults and children.[23] This indicates that movement behaviour should be considered as a whole instead of focusing on PA or SB separately.

Since the SA population is at higher risk of developing diabetes and cardiovascular ill health[26–28] and as lower levels of PA and higher levels of SB contribute to increased risk of cardiometabolic ill health[2 29] including SA participants in studies merits further investigation as the findings may inform the design and development of interventions.

A recent observational study with over 18 000 children[30] identified overweight and obesity prevalence increase during the school holiday suggesting that there are major risk factors for overweight and obesity outside of school term. From a movement point of view, it can be hypothesised that the increase in overweight and obesity is partly due to less PA and increased SB. Although children spend about 20% of the calendar year in school holidays, few studies have considered movement behaviours during school holidays. It has been hypothesised that PA would decrease while SB would increase during the school holidays when compared with school terms[31 32] because the unstructured time during holidays is more likely to foster an increase in SB and a decrease in PA.[33] Only one other study based in Japan evaluated objectively measured PA during school holidays and concluded that PA was higher during the school holidays.[31]

A possible explanation for these results are parents' levels of SB and PA. Children tend to spend more time with their parents during school holidays; if their parents have high levels of SB and low levels of PA it is likely for this behaviour to be adopted by the children also.

The aim of this study was to investigate factors associated with objectively measured movement behaviour among WB and SA primary school aged children in the District of Bradford, UK, during the school terms and school holidays. Bradford is the sixth largest city in the UK and the third with the highest population of under 16-year olds.[34] It is a highly deprived city and the health outcomes of the population are considerably worse compared with the rest of the UK.[35 36]

## METHODS

Head teachers from 15 schools were approached to take part; three volunteered and were selected. Schools were selected based on the socioeconomic status (SES) and ethnicity of the children they served. Two schools had predominantly WB children (one with children of low SES and one high) and one school with predominantly SA children with low SES.

Four hundred and ninety-two children aged 6–8 years with a mean age of 7.52 years (SD=0.5) from three schools with a Bradford postcode were invited to participate in this cross-sectional study conducted between July 2016 and May 2017. One hundred and sixty-two children were consented to participate in the study by parents. The schools were attended by children living in postcodes of varying Indices of Multiple Deprivation (IMD) of between 1 and 8.

Data were collected at five time points: summer (July 2016 term; n=55 consented /152 invited), winter (January to February 2017 term and holiday; n=69/154) and spring (April to May 2017 term and holiday; n=38/186). Primary school aged children were invited to wear an Actigraph GT3X+triaxial accelerometer (Actigraph, Pensacola, Florida, USA). The monitors were distributed in each school by the lead researcher. The Actigraph was attached on an elastic band around the child's waist on the right hip. Participating children were asked to wear the accelerometer during all waking hours and overnight for at least 7 days. Children who participated in the winter and spring data collections were invited to wear the device during school term and school holidays with a 1-week to 2-week break between the two-data collection periods. Therefore, the study comprised of three groups of children according to the season of data collection: summer, winter and spring. Each child was provided with one monitor at each data collection point. Children and teachers were provided with an explanation of how the monitor worked and that they should be removed only for water activities. Written instructions were provided to parents and teachers. Prior to giving out the monitors, parents were invited to attend a presentation and discuss any concerns they might have about the study.

### Procedure and measurements

Research measurements taken by trained staff included height (cm) (wall mounted stadiometer: Seca UK, Birmingham UK), weight (kg), body composition (measured with Tanita scales TBF-300 MA, Tokyo, Japan and Tanita scales BC-418 MA, Tokyo, Japan) and waist circumference (Seca anthropomentry tape) measured just above the exposed naval. Height was recorded to the nearest 0.1 cm and 0.1 kg, with shoes and socks removed. Body mass index (BMI) values were calculated (weight $(kg)/height (m)^2$) and BMI z-scores were calculated for each participant using the British growth reference.[37] Weight categories were derived from BMI percentiles using Freeman et al[38] recommendations.

Parents completed a demographics questionnaire which included child postcode, date of birth and ethnicity. Based on each child's home postcode, the index of multiple deprivation (IMD) was determined and collapsed into three levels of SES: low (IMD1–2), medium (IMD3–5) and high (IMD >6).[39]

## Data management and processing

The accelerometer was set to record data at a sampling rate of 60 Hz. Raw accelerometer data were downloaded and collapsed into 15 s epoch files. The Evenson[40] cut-points were selected for this study because they have been reported to be among of the most accurate to estimate MVPA for children aged 6–8 years.[41] The calibration studies for these cut-off points meet the four item criteria by Freedson *et al*[42] and Welk *et al*.[43] The choice for 15 s epoch length was in line with the calibration studies for this population group age.[40] Non-wear time was defined as ≥20 min of consecutive zero counts as they are the most reported ; 10–30 min is one of the most reported non-wear periods.[44]

The minimum valid wear time for inclusion in the analysis was at least 10 hours on at least 3 week days and at least 1 weekend day (Saturday or Sunday) for any data collection. This criterion of valid data inclusion is reported to have high reliability for estimates of PA.[45] Sleep time was removed from the data each day between 23:00 and 05:00 similar to other studies for primary school aged children[46 47] but also to take into consideration that the majority of SA children in our study were Muslims who are encouraged to wake up for the dawn prayer, which during spring and summer is around 05:00.[48]

## Patient and public involvement

The research team regularly convenes a Born in Bradford Parent Governors patient and public involvement (PPI) groups (https://borninbradford.nhs.uk/about-us/our-parents-governors/). This study was born out of parental concerns that levels of PA and sedentary time are high in childhood. Parents in the group were shown possible devices that could be used to measure PA (including Actiheart, activPAL, wrist and waist worn Actigraphs). The waist worn Actigraph was their preferred measure. Parents suggested that the best way to access children for the study was via schools; the PPI groups was not involved with recruitment in the study or the study conduct.

## DATA ANALYSIS

Statistical analyses were conducted using Stata V.13.1.[49] The three outcome variables were SB, LPA and MVPA. Since the study design involves repeated continuous measures per child, we used a multivariate linear mixed-effects multilevel modelling with child as a random effect and response variables as repeated measures nested within each child. We assumed an identity covariance structure with robust standard errors (SEs) to mitigate the impact of heteroskedasticity from skewed outcome variables. We included school as a random effect, but with only three schools we found a low intraclass correlation (ICC) (SB ICC=0.001, LPA ICC=0.01 and MVPA ICC=0.01) (see table 1). School was entered as a fixed effect covariate. All models were adjusted for the following factors: wear time, ethnicity, holiday/term, sex, SES, weight status, weekend/weekday, season and school. P values of <0.05 were defined as statistically significant. Model coefficients are displayed as mean (±SD) and confidence intervals (95% CIs) in the text and tables.

**Table 1** Performance of unconditional and full models for each outcome SB, LPA and MVPA

| Models | SB | LPA | MVPA |
|---|---|---|---|
| Nested versus non-nested likelihood ratio test | 246.02 (p<0.001) | 345.72 (p<0.001) | 393.15 (p<0.001) |
| Unconditional models (school as a random effect) | | | |
| ICC | 0.001 | 0.01 | 0.01 |
| AIC | 13991.08 | 12820.21 | 11379.40 |
| BIC | 14006.24 | 12835.37 | 11394.56 |
| Log-likelihood | −6992.54 | −6407.10 | −5686.70 |
| Unconditional models (child as a random effect) | | | |
| ICC | 0.33 | 0.39 | 0.45 |
| AIC | 13745.28 | 12482.51 | 10996.79 |
| BIC | 13760.44 | 12497.67 | 11011.95 |
| Log-likelihood | −6869.64 | −6238.26 | −5495.40 |
| Full model (child as a random effect) | | | |
| ICC | 0.33 | 0.35 | 0.37 |
| AIC | 12807.72 | 12097.39 | 10950.04 |
| BIC | 12.883.53 | 12173.19 | 11025.85 |
| Log-likelihood | −6388.86 | −6033.70 | −5460.02 |

AIC, Akaike Information Criterion; BIC, Bayesian Information Criterion; ICC, intraclass correlation; LPA, light physical activity; MVPA, moderate-to-vigorous physical activity; SB, sedentary behaviour.

**Table 2** Characteristics of children whose data were included in the analyses

| Characteristic | White British N=64 (59%) | South Asian N=44 (41%) | All children N=108 |
|---|---|---|---|
| Age (years), mean (SD) | 7.52 (0.50) | 7.61 (0.49) | 7.52 (0.50) |
| Male (%) | 32 (50) | 21 (48) | 53 (49) |
| Female (%) | 32 (50) | 23 (52) | 55 (51) |
| Low SES (%) | 6 (9.38) | 35 (79.55) | 41 (38) |
| Medium SES (%) | 40 (62.50) | 9 (20.35) | 49 (45) |
| High SES (%) | 18 (28.13) | 0 (0.0) | 18 (17) |
| z-BMI, mean (SD) | 0.31 (0.98) | 0.38 (1.34) | 0.34 (1.14) |
| Normal weight (%) | 51 (80) | 32 (73) | 83 (77) |
| Overweight (%) | 9 (14) | 3 (7) | 12 (11) |
| Obese (%) | 4 (6) | 9 (20) | 13 (12) |

BMI, body mass index; SES, socioeconomic status.

## Results

Out of 492 invited children, consent and assent were available for 160 (32.5%). A further 54 children were excluded because they did not meet the inclusion/criteria for valid data, leaving 108 (67.5%, 108/160) children who had valid data included in the analyses.

Table 2 shows the profile of the children. The mean age of the 108 children was 7.5 years with 51% girls; 59% (64/108) WB and 41% (44/108) were SA. Forty-one (38%; 41/108) children were from the 20% most deprived areas (IMD 1–2), however, that was the case for the majority of SA (80%; 35/44) children. About three-quarters of children (77%, 83/108) were of normal weight and 23% (25/108) were overweight and obese although a higher percentage (27%; 12/44) of SA than WB children were overweight and obese (table 2).

When we compared the characteristics of the children whose data were included in the analyses to those excluded (n=54), we did not find any differences relating to age, sex, SES or overweight status.

The 108 children who had valid data provided a total of 1157 term days and holiday days. Children (SA and WB combined) had on average 901 min of daily valid wear of which 539 min (60%) were spent SB, 296 min (33%) in LPA and 67 min (7%) were MVPA.

Table 3 gives an overview of the unadjusted wear times, SB, LPA and MVPA average minutes stratified by ethnicity, school terms and school holidays. The unadjusted results in the combined data set identified slightly higher proportions of SB (61%; 550 min) and LPA (33%; 292 min) but lower proportions of MVPA (6%; 57 min) for SA children compared with WB children. A similar pattern was observed when stratifying the results by term and holiday (table 3).

**Table 3** Unadjusted data sets averages by ethnicity

| Combined data set | Combined data | | | | School terms | | | | School holidays | | | |
|---|---|---|---|---|---|---|---|---|---|---|---|---|
| | WB N=64 Mean (SD) | % of wear time | SA N=34 Mean (SD) | % of wear time | WB N=56 Mean (SD) | % of wear time | SA N=39 Mean (SD) | % of wear time | WB N=40 Mean (SD) | % of wear time | SA N=23 Mean (SD) | % of wear time |
| Total number of valid days | 684 | n/a | 473 | – | 369 | n/a | 269 | – | 315 | n/a | 214 | – |
| Valid wear time in minutes | 888.78 (117.16) | n/a | 899.85 (110.55) | – | 876.66 (105.60) | n/a | 902.12 (120.55) | – | 885.88 (105.83) | n/a | 915.03 (85.22) | – |
| Total SB in minutes | 532.05 (109.13) | 60 | 550.13 (117.57) | 61 | 532.42 (104.44) | 61 | 553.05 (113.56) | 61 | 522.10 (110.2) | 60 | 564.55 (76.89) | 61 |
| Total LPA | 288.73 (66.64) | 32 | 292.27 (52.02) | 33 | 275.54 (64.78) | 31 | 289.27 (55.03) | 32 | 299.90 (67.14) | 33 | 296.90 (56.47) | 32 |
| Total MVPA | 67.99 (33) | 8 | 57.44 (37.97) | 6 | 68.95 (31.49) | 8 | 59.80 (34.08) | 7 | 63.85 (31.44) | 7 | 53.57 (24.88) | 7 |

LPA, light physical activity; MVPA, moderate-to-vigorous physical activity; SB, sedentary behaviour.

**Table 4** Model coefficients for three movement behaviour outcomes based on the combined data set (term and holiday days; n=108) (95% CI)

| Covariates | SB Coefficient, CI, p value | LPA Coefficient, CI, p value | MVPA Coefficient, CI, p value |
|---|---|---|---|
| Total wear time | 0.68 (0.64 to 0.71) p<0.001 | 0.28 (0.25 to 0.31) p<0.001 | 0.05 (0.03 to 0.06) p<0.001 |
| South Asian | 38.51 (6.79 to 70.23) p=0.017 | −23.66 (−47.46 to 0.15) p=0.051 | −14.94 (−34.41 to 4.54) p=0.133 |
| Male | −1.15 (−18.88 to 16.59) p=0.899 | −9.87 (−23.08 to 3.33) p=0.143 | 11.03 (2.33 to 19.72) p=0.013 |
| Medium SES (ref low SES) | 15.82 (−6.09 to 37.72) p=0.157 | −17.29 (−31.65 to −2.92) p=0.018 | 1.65 (−12 to 15.3) p=0.813 |
| High SES (ref low SES) | 43.26 (12.35 to 74.18) p=0.006 | −31.28 (−55.36 to −7.2) p=0.011 | −11.85 (−27.61 to 3.9) p=0.14 |
| School A (ref school B) | 18 (−23.86 to 59.86) p=0.399 | −14.01 (−45.87 to 17.85) p=0.389 | −4.31 (−28.37 to 19.74) p=0.725 |
| School C (ref school B) | 18.64 (−14.36 to 51.65) p=0.268 | −11.37 (−36.68 to 13.95) p=0.379 | −7.36 (−27.06 to 12.35) p=0.464 |
| Spring (ref winter) | −53.48 (−81.56 to −25.4) p<0.001 | 38.07 (17 to 59.13) p<0.001 | 15.48 (0 to 30.97) p=0.05 |
| Summer (ref winter) | −42.44 (−59.61 to −25.28) p<0.001 | 26.99 (13.28 to 40.7) p<0.001 | 15.47 (7.15 to 23.8) p<0.001 |
| Term (ref holiday) | 8.04 (−1.38 to 17.46) p=0.094 | −8.64 (−15.38 to −1.9) p=0.012 | 0.59 (−3.84 to 5.01) p=0.796 |
| Weekend (ref weekday) | −6.02 (−15.44 to 3.4) p=0.211 | 1.39 (−5.66 to 8.43) p=0.7 | 4.67 (0.31 to 9.02) p=0.036 |
| Overweight and obese (ref normal weight) | −19.18 (−42.03 to 3.68) p=0.1 | 21.13 (3.23 to 39.04) p=0.021 | −1.94 (−11.25 to 7.37) p=0.683 |
| Intercept | −86.66 (−133.21 to −40.1) p<0.001 | 64.81 (29.53 to 100.09) p<0.001 | 21.7 (−1.16 to 44.55) p=0.063 |
| Variance of level-2 (child) | 147.11 | 915.14 | 357.36 |
| ICC | 0.33 | 0.35 | 0.37 |

CI, Confidence Interval; ICC, intraclass correlation; LPA, Light physical activity; MVPA, moderate-to-vigorous physical activity; SB, Sedentary behaviour.

Table 4 reports findings from statistical models for each outcome variable: SB, LPA and MVPA following mutual adjustment for predictors of interest. Once wear time, term, ethnicity, SES, school, season, sex, weekend and weight status were mutually adjusted for, significant differences were identified for SB in relation to three factors: ethnicity, SES and seasonality; for LPA in relation to four factors: SES, seasonality, term and overweight; and for MVPA in relation to three factors: sex, seasonality and weekend (table 4).

Specifically, boys spent significantly more time in MVPA (11 min/day) compared with girls and SA children spent more time in SB (39 min) compared with WB children (p=0.017). Children residing in high SES area had ~43 min/day more sedentary time and 31 min/day less LPA than children from low SES areas (p<0.001; p=0.011).

Children were more active (15 min MVPA, 27 min LPA) and less sedentary (−42 min) during summer and spring (15 min MVPA, 38 min LPA, −53 min SB) compared with winter (p<0.001). Less time in LPA (9 min) during school terms compared with school holidays (p=0.012). Children spent more time in MVPA (5 min) during weekends compared with weekdays (p=0.036). Overweight and obese children spent more time in LPA (21 min) than normal weight children (p=0.021).

## DISCUSSION
### Summary of findings
Overall, children spent a large proportion of time in SB, ~60%–61% (522–564 min/day), followed by LPA, ~31%–33% (276–300 min) and a smaller proportion in MVPA, 6%–8% (54–69 min). We also identified other key findings: (1) SA children were more sedentary than WB children; (2) boys were more active than girls; (3) there were no significant differences for SB and MVPA, yet LPA was lower during term compared with school holiday; (4) children had more MVPA during weekends compared with weekdays; (5) children were markedly more active and less sedentary during the summer and spring compared with winter; (6) overweight and obese children has more LPA than normal weight children and (7) children from high SES were more sedentary and less active (LPA) compared with children from low SES.

### Comparison to other studies
We found that SA children spent 39 more min/day in SB compared with WB children which is similar to the results of other studies evaluating ethnic differences in PA and SB in WB and SA, children aged 8–9 years[50] and girls aged 9–11 years in England.[10] Both studies found that SA children were less active and more sedentary compared with WB children. The results of the Millennium Study also reported that SA children were the most sedentary and least active when compared with WB or children of African origin.[51]

One possible explanation for higher SB in SA children is Mosque attendance for religious education and Arabic language. Data from 2356 children of SA ethnicity from the Born in Bradford cohort study identified that 71% of SA children attended the religious setting most days of the week and spent 1–2 hours here. It is likely that the time spent here is seated and this might explain higher sedentary time for SA children. Furthermore, it has previously been shown that SA children watch more TV from a younger age[9] and sleep less overnight allowing more time for SB.[52] Results from the Born in Bradford study have also shown that both sleep and TV viewing are also related to fatness/obesity in Bradford children (higher in SA than in WB) which is inversely associated with PA levels.[8]

Overall, children spent a large proportion of time in SB, followed by LPA, and a smaller proportion in MVPA, 6%–8% (57–69 min) (table 3). Although the children

in our study had high levels of SB, the average level of MVPA was higher than compared with other studies.[51 53] Results from the Millennium Study[51] on 13 000, children aged 7 years identified an average of 38 min/day spent in MVPA. The Millennium Study, however, had a different definition of a valid day and included days with >6 hours or valid wear time. It is possible that the higher levels of MVPA in our study are related data being collected over 3 weeks of summer and spring and only 2 weeks of winter.

There were only modest differences between the school term and the school holidays, with 8.64 fewer min/day of LPA spent during the school term time compared with the school holidays. Our finding lower LPA and no differences for SB and MVPA during the school term is contrary to previous reports of higher SB and lower PA during school holidays.[54] It is possible that children maintain the movement habits from the school term during the school holidays despite the lack of unstructured time for some and despite not needing to sit down for lessons. Currently, the majority of interventions aimed at increasing PA and/or reducing SB focus on school term time only.[55–57] Our results imply that interventions to promote positive movement behaviour are needed also during holidays as high levels of SB have been recorded during both occasions. From a practical point of view for data collection and interventions, children are more accessible en-masse during the school term; however, the school holiday period should be considered as it appears to be a critical time period for some children. In a recent meta-analysis[58] which highlighted reasons for the lack of effectiveness of PA interventions, it was suggested that the positive outcomes of PA interventions during school hours are not maintained outside of school hours and that interventions should include homes and communities.

Seasonality was identified as a consistent correlate of all movement behaviours (SB lower and LPA, MVPA higher during the summer and spring season compared with winter). These patterns are similar to the study of Atkin et al[59] which identified elevated levels of SB in winter and autumn for children aged 7 years and suggested the need for interventions during autumn and winter. Similarly, a study involving over 1000 primary school aged Danish children,[53] found higher SB and lower MVPA during winter compared with spring. More studies should consider designs that allow seasonal comparisons within the same children measured repeatedly throughout the course of a year. Regarding weekdays and weekend, unlike our study, Hjorth et al[53] identified higher SB and lower MVPA at the weekend. Most studies evaluating physical evaluating movement behaviour in children[14 19 60] have identified lower MVPA at the weekend which is contrary to our results.

Compared with children from low SES areas, children residing in higher SES areas (IMD 6–8) were more sedentary and less active which is contrary to majority of the existing evidence. A possible explanation would be that children residing in higher SES area might have access to more technology devices and engage in more screen

viewing activities. Also, contrary to existing evidence,[8 52] the overweight and obese children were more active than normal weight children.

## Strengths and limitations

The strength our study is related to the uniqueness of the sample which included SA and WB children from SES areas with IMD 1–8, and data during the school holidays. To our knowledge, this is the first accelerometry study in Europe to take into consideration school holidays.

The 33% response rate in our study is similar to response rates in similar studies taking part in more affluent areas in the UK.[14 61] The sample of the study was modest which means that we may have been underpowered to detect small associations. However, our sample size was similar to a Japanese study conducted by Tanaka and colleagues which considered the summer holidays accelerometry and included 98 children in the data analysis.[31] A larger study which considered PA and SB during school holidays was conducted in the USA and included 406 children but the data were collected over 20 months and PA and SB were self-reported.[32]

Despite aiming to include children from various SES areas, this was challenging because the majority of SA children in Bradford were from the poorest areas with only 20% of SA children from medium SES and 0% from high SES. The use of area SES has its own limitations by introducing misclassification error.[62]

Ideally, a comparator school with predominantly SA children with high SES would also have been included, but the demographics of Bradford precluded this.

Another limitation of the study could be the definition of sleep and the 05:00 awake time. It is possible that some sleeping hours may have been misclassified as SB, thereby overestimating SB. Our choice for the 05:00 awake time was made in the context of SA children, majority of Muslim faith, who are encouraged to wake up around this time for the morning prayer.

## CONCLUSION

The results of our study suggest that children spent high proportions of their time in SB. Factors associated with movement behaviours included ethnicity, sex, weight status, area level SES and temporal factors: weekend, school holidays and seasonality. SA children were more sedentary and less active than WB children. Interventions to reduce SB and increase PA are needed over school term and school holidays. Interventions aimed at reducing SB and increasing PA are needed across all seasons but especially during the winter when SB is higher and PA is lower.

**Acknowledgements** We thank all the participants and their families for their time and ongoing participation in the study. We also thank all the researchers involved in data collection.

**Contributors** Study design and research question: LCN, MH, SB, MM and PC. Data collection: LCN, MH and PC. Data processing and analysis: all authors contributed to discussions on data analysis. Data processing and analysis was carried out by LCN, MM and MF. Ongoing discussion and comments: all. Drafted paper and final paper: LCN.

**Funding** This research was funded by the NIHR Collaboration for Leadership in Applied Health Research and Care Yorkshire and Humber (NIHR CLAHRC YH). www.clahrc-yh.nihr.ac.uk. PC is funded by a BHF Immediate Postdoctoral Basic Science Research Fellowship (FS/17/37/32937).

**Disclaimer** The views and opinions expressed are those of the authors, and not necessarily those of the NHS, the NIHR or the Department of Health, or any other funding agency.

**Competing interests** None declared.

**Patient consent for publication** Obtained.

**Ethics approval** Ethical approval for the study was obtained from the University of Bradford Ethical Committee (E536, 2016).

**Provenance and peer review** Not commissioned; externally peer reviewed.

**Data availability statement** No extra data are available. All data relevant to the study are included in the article or uploaded as supplementary information.

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
