## [Reviewer comments · BMJ Open]

ARTICLE DETAILS

TITLE (PROVISIONAL)	Factors associated with accelerometer measured movement behaviours among 6-to-8-year-old White British and South Asian children during school terms and school holidays
AUTHORS	Nagy, Liana; Faisal, Muhammad; Horne, Maria; Collings, Paul; Barber, Sally; Mohammed, Mohammed

VERSION 1 - REVIEW

REVIEWER	Yeonwoo Kim School of Kinesiology and Institute for Social Research, University of Michigan, USA
REVIEW RETURNED	24-Aug-2018

GENERAL COMMENTS	This study examined factors of movement behaviors among White British and South Asian children during school terms and holidays. This study showed that ethnicity, area-level SES, seasonality, overweight, and weekend were associated with at least one type of physical activity outcomes. This study makes a useful contribution to the literature by including White British and South Asian children and examining school holidays as well as school terms. However, I note several issues that require further clarification, consideration, or revision. 1. Literature supporting the hypothesis: In the Introduction section, the authors elaborated the need to include South Asian children in studies and describe both school terms and holidays. It seems to me that the authors focus on ethnicity and school term as correlates of physical activity. Please provide conceptual/theoretical reasons supporting ethnic differences in physical activity between South Asian children and White British children. I also suggest the authors to provide more detail here about how school terms theoretically affect child physical activity and sedentary behaviors.2. The data has a low response rate. Out of 492 invited children, only 160 (or 162?) children participated in this study. Then, 54 out of 160 children were excluded because of invalid data. Please address a low response rate in the limitation section. Also, they need to compare sample characteristics between those who were excluded (n=54) and those who were included (n=106).
--

	3. Multilevel modeling: More information would help readers understand multilevel models in this paper. I suggest the authors to specify the following information. (1) A reason why multilevel modeling is required in this study - findings of an unconditional model, ICC, and data clustering. (2) Model selection process: Steps in the construction of the models need to be included in text. Please specify model fits for each model and how the authors determined the best-fitting multilevel model among non-nested models. (3) The level-1, level-2, and combined equations of the final model: Equations of the final model would improve understanding of statistical models. (4) Values of intercept and level-2 variance in Table 3. Specific Comments 1. The number of respondents is 160 in abstract, but 162 in Methods (p. 4). 2. Please describe the process of selecting schools. 3. Please include both parent's education and income in the analytic models because these SES factors have been found to be associated with children's physical activity. 4. Please specify that the IMD and SES in the Tables were measured at the area level, not at the family level. Also, it is hard to say that the authors included children from various family-level SES because children living in low SES areas are not meant to be those in low-income families. 5. Please elaborate how missing data were handled in the Methods section.
--	--

REVIEWER	Bruno Gonçalves Galdino da Costa Federal University of Santa Catarina, Brazil
REVIEW RETURNED	13-Sep-2018

GENERAL COMMENTS	Review Manuscript: Factors associated with accelerometer measured movement behaviours among 6-to-8-year-old White British and South Asian children during school terms and school holiday Dear editor, This article is well-written, and its methods are appropriate to answer its objective. Some details regarding the sampling, the accelerometer protocol and the statistical analyses could be better clarified. The results are interesting, revealing many correlates of sedentary behaviours, and of light- and moderate-to-vigorous-intensity physical activities. In relation to the discussion, it lacks explanation to the findings of the study. The authors compared their results with other studies, but it is not discussed why the associations found were found and neither their implications for the participants and for future studies. Interpreting and explaining the findings may improve this section.
--

My comments on each section of the manuscript are provided below. I hope these suggestions will help improve the quality of this article.

Abstract and Summary

The abstract is clear and concise, the methods and results are well described, and the message is adequately written in the conclusion section.

a) It should be clearly stated that three samples were analyzed in the participants subsection of the abstract.

b) My second concern is in the first sentence of the methods section of the abstract. Describing the number of children and number of valid days is not very informative, it reads vaguely. It would be preferred if the proportion of children that provided enough valid data to be included in the analysis was described. Another possibility would be reporting the average number of valid days with accelerometer data each participant provided.

Introduction

The introduction section is also well-written. I have only three suggestions:

a) On page 3, lines 40-55, the authors describe ethnical differences in the risk of developing diseases and ethnical differences between levels of MVPA and SB, but no differences related to the correlates and determinants of MVPA and SB. Why would these differences in diseases and behaviours exist? Are there any hypotheses or possible explanations for these differences? Perhaps some environmental, cultural or economic variables should be considered here since they are related to movement behaviours and might differ between the ethnical groups analyzed[1].

b) On page 4, lines 1-7, the authors rationalize that children would engage in less PA and more SB during holidays compared to school terms due to the unstructured characteristic of this period. Although their reasoning is backed by references, my suggestion is that the authors should explain this difference further. Students do not engage in school-related desk-bound activities during holidays, which they cannot opt not to do during school terms, when this may account for a large proportion of children's daily volume of SB [2–5]. On the other hand, during holidays they do not have physical education classes or school-related extracurricular sports that contribute to daily PA levels. In conclusion, what differences in daily activities result in more SB and less PA during holidays when compared to school days?

c) Explicitly writing the variables MVPA, LPA and SB, instead of using the term "movement behaviour" on the objective may make it clearer for the readers.

Methods

The procedures and measurements are well described. However, some procedures can be further explained.

a) Why were these three schools invited? Were they selected using any specific criteria?

b) In relation to the accelerometry protocol, some information could be better described. Examples are: how many monitors were distributed? How were they distributed? The researchers explained how it worked, and if it should be removed for water activities or for

sleeping? Although this study is not an intervention study, Montoye et al. provided a checklist that may help to report the accelerometry protocol. [6]

c) The authors describe a mixed-effects linear regression analysis adjusted by variables of interest, however, it is not clear what variables are on the random or fixed effects of the model.

Considering that the analyzed sample consisted of three different samples that used the accelerometer according to the seasons, authors should describe how they were handled in the final model. In addition, the authors could check the models for heteroscedasticity and collinearity and evaluate or report the model parameters for multilevel analyses (e.g. AIC, BIC) if they have already evaluated them.

Results

a) The number of participants on each model could be displayed in table 3. In addition, on the description of the table it is stated that model coefficients and confidence intervals are reported, but on the table, the values written are means and min and max values. These terms could be written in a standardized way to prevent ambiguous interpretation.

b) The reference categories in table 3 should be explicitly indicated.

Discussion

a) On the first two lines of the discussion section, the authors summarize the results. However, the values for SB are in the range of 550-564min/d (page 10, line 53), whereas on the results section, it is stated that participants spent an average of 539 min/d on SB (page 6, line 48), which is not included on the aforementioned range. These and the other values provided should be checked and clarified so the reader will know from which results your conclusions are being drawn from.

b) Although the authors compare their results with other studies with similar design, the findings are not explained. On page 11, lines 11-19, the authors conclude that the evidence up to date shows that SA children are less active and more sedentary, yet, there is no possible explanation to why this is observed. The same structure can be observed in the following paragraphs, where the differences between holidays and terms and between seasons are highlighted, but no explanations are proposed or hypothesized.

c) In relation to the limitation due to considering waking time at 5 am, some authors have proposed a way to identify sleep and SB from accelerometer data worn on the waist⁷. Maybe this can give some insight into how to improve the accuracy to identify sleep from SB, considering that 5 am is rather early.

References

1. Bauman, A. E. et al. Correlates of physical activity: why are some people physically active and others not? *Lancet* 380, 258–71 (2012).
2. Bailey, D. P. et al. Accelerometry-assessed sedentary behaviour and physical activity levels during the segmented school day in 10-14-year-old children: the HAPPY study. *Eur J Pediatr* 171, 1805–13 (2012).
3. Nettlefold, L. et al. The challenge of low physical activity during the school day: at recess, lunch and in physical education. *Br J Sports Med* 45, 813–9 (2011).
4. Guinhouya, B. C. et al. How school time physical activity is the 'big one' for daily activity among schoolchildren: a semi-experimental approach. *J Phys Act Health* 6, 510–9 (2009).

	5. da Costa, B. G. G. et al. Sociodemographic, biological, and psychosocial correlates of light-and moderate-to-vigorous-intensity physical activity during school time, recesses, and physical education classes. J. Sport Health Sci. doi:10.1016/j.jshs.2017.05.002 6. Montoye, A. H. K., Moore, R. W., Bowles, H. R., Korycinski, R. & Pfeiffer, K. A. Reporting accelerometer methods in physical activity intervention studies: a systematic review and recommendations for authors. Br J Sports Med bjsports-2015-095947 (2016). doi:10.1136/bjsports-2015-095947 7. Tudor-Locke, C. et al. Nocturnal sleep-related variables from 24-h free-living waist-worn accelerometry: International Study of Childhood Obesity, Lifestyle and the Environment. Int. J. Obes. Suppl. 5, S47–S52 (2015).
--	---

REVIEWER	Xiangli Gu University of Texas at Arlington
REVIEW RETURNED	02-Nov-2018

GENERAL COMMENTS	I think authors should clearly define the "movement behavior" and the introduction should be reorganized with more literature review. The purpose of the study is not clear to the reader. In addition to this, the method especially the data analysis is a major concern of the paper. The authors used linear mixed effects model to analyze the relationship between three continuous outcome variables and both school- and child-level variables. I have the following concerns and comments that need the authors to clarify.  1) There was no explanation about the linear model effects model form, including the fixed and random effects and the structure of the covariance of the random effects and errors. Please add it. 2) How many levels are included in the analysis? What are the variables at each level? What does the variable "School" refer to? 3) It is unclear how well the model fits the data as compared to unconditional means model and unconditional growth or random coefficient model. Please use fit statistics (e.g., deviance or likelihood ratio test) to compare the models and justify that the selected model has the best fit. 4) It is necessary to report the variances at different levels as well as the intra-cluster correlation to show the amount of the total variance explained by variables at each level and the extent of clustering. 5) The presentation of the result in Table 3 is confusing. First, a typical journal table format does not use borders inside the table. Second, in the main text, the authors define 3 categories of school terms: winter, summer and spring. The table shows that the spring term is chosen as the reference category. Why is "school term" variable included as a different variable? The authors should specify the reference category for each categorical variable and present all non-reference categories under this same categorical variable. Please work on the discussion to consistent with your research questions as well as the terms used throughout the paper Thank you.
---

VERSION 1 – AUTHOR RESPONSE

Reviewer 1	Author's comments
1. Literature supporting the hypothesis: In the Introduction section, the authors elaborated the need to include South Asian children in studies and describe both school terms and holidays. It seems to me that the authors focus on ethnicity and school term as correlates of physical activity. Please provide conceptual/theoretical reasons supporting ethnic differences in physical activity between South Asian children and White British children. I also suggest the authors to provide more detail here about how school terms theoretically affect child physical activity and sedentary behaviors.	Thank you for your comments. We have highlighted on page 3 and 4 that most of existing research indicates higher levels of SB and lower levels of PA in South Asian children when compared to White British. We have included a section on page 3 specifying that the South Asian population have a higher risk of developing diabetes and cardiovascular illness. We have added that since low levels of PA and high levels of SB contribute to increased risk of cardiometabolic ill health, including South Asian participants in studies merits further investigation so that appropriate interventions can be developed. We have included both school terms and school holidays in the study and we have now highlighted the existing hypothesis that PA would decrease, and SB would increase on the basis of unstructured time during the holidays (page 4). To date there are no European studies reporting accelerometry data during school holidays and we are not aware of other existing hypotheses in the research literature other that relating to increase in BMI during school holidays. We have added a section to strengthen our hypothesis and referred to a US a study (2016) on over 18,000 kindergarten and primary school children which showed increased of overweight and obesity during the school holidays compared to school term.
2. The data has a low response rate. Out of 492 invited children, only 160 (or 162?) children participated in this study. Then, 54 out of 160 children were excluded because of invalid data. Please address a low response rate in the limitation section. Also, they need to compare sample characteristics between those who were excluded (n=54) and those who were included (n=106).	A response rate for 33% is not unusual for the population of Bradford or even more affluent and educated populations (ROOTS study and EPIC-Norfolk). We have added this references in the strength and limitation section on page 13. A similar response rate was identified in other studies involving accelerometer data in the Born in Bradford project. 162 children had parent consent but on the day of data collection 2 children were absent, therefore on 160 provided child assent. This is clarified on page 6. We used a strict criteria for valid data inclusion which has contributed to the loss of 33% (n=54) of the data. We compared the characteristics of included (n=108) and excluded children and did not find any differences relating to age, sex, SES, or

	overweight status. We have included this information on page 7 just above table 2.
3. Multilevel modeling: More information would help readers understand multilevel models in this paper. I suggest the authors to specify the following information. (1) A reason why multilevel modeling is required in this study - findings of an unconditional model, ICC, and data clustering. (2) Model selection process: Steps in the construction of the models need to be included in text. Please specify model fits for each model and how the authors determined the best-fitting multilevel model among non-nested models. (3) The level-1, level-2, and combined equations of the final model: Equations of the final model would improve understanding of statistical models. (4) Values of intercept and level-2 variance in Table 3.	The choice for the multilevel modelling is due to repeated measurement within child. We included school as a random effect, but there are only three schools and ICC is low for all the models (SB ICC=0.001, LPA ICC=0.01, and MVPA ICC = 0.01) (see Table 1 on page 6). So, we have used child as a random effect and school as a fixed effect variable. Further, we have now clarified that we are using identity covariance structure (page 6 – Data analysis). We have two covariates at level 1 (repeated measures) that is wear time and school term, and at level 2 (child) we considered all remaining variables (e.g., ethnicity, gender, SES, Season, Weekend, and BMI). We have now reported performance measures (i.e., AIC, BIC, LR test) for all models (see Table 1- page 6). We have now included this explanation in the manuscript We have now also added the values of intercept and level-2 variance in Table 4 (previously table 3).
4. The number of respondents is 160 in abstract, but 162 in Methods (p. 4).	Although 162 children had parent consent, only 160 provided child assent. For clarity we have changed the number in the results section to 160 (page 7). The 162 children in the methods section specifies parent consent only.
5. Please describe the process of selecting schools.	The schools were selected based on ethnicity (to include White British and South Asian children) and SES (to include high and low SES) as specified on page 4. Although we approached 15 schools for participation, only 3 schools volunteered to participate. We have now included this on page 4 in the methods section.
6. Please include both parent's education and income in the analytic models because these SES factors have been found to be associated with children's physical activity.	Unfortunately, these personal level data were not available. We have included the area SES as a limitation of our study on page 14.
7. Please specify that the IMD and SES in the Tables were measured at the area level, not at the family level. Also, it is hard to say that the authors included children from various family-level SES because children living in low SES areas are not meant to be those in low-income families.	Child postcode was used to generate the IMD, this has now been made clear on page 5; a sentence has been added to page 4 for further clarity. We have acknowledged the limitations of using area SES on page 14.

8. Please elaborate how missing data were handled in the Methods section.	We had some missing postcodes for four children and we were unable to trace them as the children have moved schools by the time we've returned to the schools to request info on any missing data. We have therefore allocated 2 children for each SES group based on the various IMD levels. There was an ongoing exiting liaison between the schools and the lead researcher and efforts were made to recover any missing data.
Reviewer 2	
1. It should be clearly stated that three samples were analyzed in the participants subsection of the abstract.	Thank you for your comment. Quantitative data from all data collections points were merged into one sample. It included children from three schools with data collections during three seasons and during school terms and school holidays. We have made amendments to the abstract to clarify this.
2. My second concern is in the first sentence of the methods section of the abstract. Describing the number of children and number of valid days is not very informative, it reads vaguely. It would be preferred if the proportion of children that provided enough valid data to be included in the analysis was described. Another possibility would be reporting the average number of valid days with accelerometer data each participant provided.	67.5 % (108/160) children provided valid data - we have added this in the abstract. We feel that including the total number of valid days provides the reader with an idea of the size of the data set. A data set of 108 children could generate a data set of at least 108 days but in this data was collected during both school term and holidays generating 1157 valid days.
3. On page 3, lines 40-55, the authors describe ethnic differences in the risk of developing diseases and ethnic differences between levels of MVPA and SB, but no differences related to the correlates and determinants of MVPA and SB. Why would these differences in diseases and behaviours exist? Are there any hypotheses or possible explanations for these differences? Perhaps some environmental, cultural or economic variables should be considered here since they are related to movement behaviours and might differ between the ethnic groups analyzed [1].	The differences in disease and behaviours between South Asians and White British are well established. We have addressed this question above in more detail, in the comments to Questions 1 (reviewer 1). We have controlled for SES status using the Index of Multiple Deprivation, but we have not collected any specific data on environment or culture to be able to discuss it. Other factors explored by our study is sex, seasonality, day of the week and overweight status.
4. On page 4, lines 1-7, the authors rationalize that children would engage in less PA and more SB during holidays compared to school terms due to the unstructured characteristic of this period. Although their reasoning is backed by references, my suggestion is that the authors should explain this difference further. Students do not engage in school-related desk-bound activities during holidays, which they cannot opt not to do during school terms, when this may account for a large proportion of children's daily	Although students might not engage in desk activities it is likely they engage in more screen viewing activities (TV, computer games, some social media). Spending more time with parents who are not physically active during the school holiday can be another contributor reduced PA and increased SB. We have added this on page 4.

volume of SB [2–5]. On the other hand, during holidays they do not have physical education classes or school-related extracurricular sports that contribute to daily PA levels. In conclusion, what differences in daily activities result in more SB and less PA during holidays when compared to school days?	We have not talked about differences in daily activities because we do not have this type of data available.						
5. Explicitly writing the variables MVPA, LPA and SB, instead of using the term “movement behaviour” on the objective may make it clearer for the readers.	The term movement behaviour was used to reflect current approaches in MVPA, LPA and SB research where behaviours are no longer considered in isolation, but on a spectrum of energy expenditure. Pedesic (2017) Chastin (2018) support this view.						
The procedures and measurements are well described. However, some procedures can be further explained. 6. Why were these three schools invited? Were they selected using any specific criteria?	Schools in Bradford tend to be predominantly South Asian, or predominantly White British. We identified two predominantly White British schools (high and low SES areas) and one South Asian school (low SES). Ideally, we would have liked a South Asian school from a high SES area but because of Bradford demographics we were not able to identify one. This has been clarified in the methods section on page 4 and results section, page 7.						
7. In relation to the accelerometry protocol, some information could be better described. Examples are: how many monitors were distributed? How were they distributed? The researchers explained how it worked, and if it should be removed for water activities or for sleeping? Although this study is not an intervention study, Montoye et al. provided a checklist that may help to report the accelerometry protocol. [6]	Each child was provided with one monitor at each data collection. The monitors were distributed in schools by the main researcher. Children and teachers were explained how the device worked and that it should be removed only for water activities. Parents were also invited to attend a presentation and discuss any concerns they might have about the study. Only some attended. We have included this information on page 4.						
8. The authors describe a mixed-effects linear regression analysis adjusted by variables of interest, however, it is not clear what variables are on the random or fixed effects of the model. Considering that the analyzed sample consisted of three different samples that used the accelerometer according to the seasons, authors should describe how they were handled in the final model. In addition, the authors could check the models for heteroscedasticity and collinearity and evaluate or report the model parameters for multilevel analyses (e.g. AIC, BIC) if they have already evaluated them.	We have now clarified that the child is a random effect. We included school as a random effect, but there are only three schools and ICC is low for all the models (SB ICC=0.001, LPA ICC=0.01, and MVPA ICC = 0.01) (see Table 1). Therefore, we included only child as a random effect. We have specified on page 6 that we have adjusted for season (as fixed effect) in the final model. We have now described that we used the robust standard error that mitigates the impact of heteroscedasticity. We have also evaluated the multicollinearity and we found mean variance inflation factor (VIF) =2.24 and individual covariate VIF<6, which is acceptable.    Variable VIF 1/VIF     totalweartime 1.04 0.959875   	Variable	VIF	1/VIF	totalweartime	1.04	0.959875
Variable	VIF	1/VIF					
totalweartime	1.04	0.959875					

	South Asian	4.4	0.227233
	Male	1.13	0.886399
	imd		
	Medium SES	2.6	0.384602
	High SES	2.64	0.37935
	sch		
	A	5.75	0.17399
	C	3.47	0.288098
	season_code		
	Spring	1.11	0.903744
	Summer	1.3	0.767619
	Term	1.29	0.777094
	Weekend	1.01	0.992665
	Overweight	1.11	0.902658
	Mean VIF	2.24	
Results 9. The number of participants on each model could be displayed in table 3. In addition, on the description of the table it is stated that model coefficients and confidence intervals are reported, but on the table, the values written are means and min and max values. These terms could be written in a standardized way to prevent ambiguous interpretation.	There are 108 participants with valid data discussed throughout the article. we have added n=108 in the title of Table 4(previously table 3).Each model has 108 children.		
10. The reference categories in table 3 should be explicitly indicated.	The categories using the terms SB, LPA and MVPA have been given in full in the introduction section. We have now added them again at the end of table 4(previously table 3).		
Discussion 11. On the first two lines of the discussion section, the authors summarize the results. However, the values for SB are in the range of 550-564min/d (page 10, line 53), whereas on the results section, it is stated that participants spent an average of 539 min/d on SB (page 6, line 48), which is not included on the aforementioned range. These and the other values provided should be checked and clarified so the reader will know from which results your conclusions are being drawn from.	Thank you for highlighting this. We have amended the correct range on page 10(now page) , which as reported in Table 3 (previously table 2 at the first submission) was between 522 to 564 min/d.		
12. Although the authors compare their results with other studies with similar design, the findings are not explained. On page 11, lines 11-19, the authors conclude that the evidence up to date shows that SA children are less active and more sedentary, yet, there is no possible explanation to why this is observed. The same structure can be observed in the	We have added our hypotheses on page 12: "One possible explanation for higher SB in SA children is Mosque attendance on a daily basis for religious education and Arabic language. Data from 2356 children of SA ethnicity taking part in the Born in Bradford cohort study identified that 71 % of SA children attended the religious setting most days of the week and		

following paragraphs, where the differences between holidays and terms and between seasons are highlighted, but no explanations are proposed or hypothesized.	spent 1-2 hours here. It is likely that the time spent here is seated and this might explain higher sedentary time for SA children. Furthermore, it has previously been shown that SA children watch more TV from a young age (9) and sleep less overnight allowing more time for SB (51). Results from the Born in Bradford study have also shown that both sleep and TV-viewing are also related to fatness/ obesity in Bradford children (higher in SA than in WB) which is inversely associated with PA levels (8).”
13. In relation to the limitation due to considering waking time at 5 am, some authors have proposed a way to identify sleep and SB from accelerometer data worn on the waist⁷. Maybe this can give some insight into how to improve the accuracy to identify sleep from SB, considering that 5 am is rather early.	Thank you for your comment. We acknowledge this as a limitation on page 14 and specified that the reason for choosing 5 am relates to a high percentage of Muslim children, who are encouraged to wake up early for prayer. This approach is not dissimilar to other accelometry studies which we have referenced on page 6.
Reviewer 3	
1. There was no explanation about the linear model effects model form, including the fixed and random effects and the structure of the covariance of the random effects and errors. Please add it.	The choice for the multilevel modelling is because we have repeated measurement within child. We included school as a random effect, but there are only three schools and ICC is low for all the models (SB ICC=0.001, LPA ICC=0.01, and MVPA ICC = 0.01) (see Table 1- data analysis section). So, we have used child as a random effect and school as a fixed effect variable. Further, we have now clarified that we are using identity covariance structure. We have two covariates at level 1 that is wear time and school term and at level 2 we considered all remaining variables (e.g., ethnicity, gender, SES, Season, Weekend, and BMI). We have now reported performance measures (i.e., AIC, BIC, LR test) for all models (see Table 1 added in data analysis section). We have now also included this explanation in the manuscript on page 6.
2. How many levels are included in the analysis? What are the variables at each level? What does the variable “School” refer to?	
3. It is unclear how well the model fits the data as compared to unconditional means model and unconditional growth or random coefficient model. Please use fit statistics (e.g., deviance or likelihood ratio test) to compare the models and justify that the selected model has the best fit.	We have now reported fit statistics for unconditional means and full models (see Table 1).
4. It is necessary to report the variances at different levels as well as the intra-cluster correlation to show the amount of the total variance explained by variables at each level and the extent of clustering.	We have initially not reported variances because the aim of your study is not to explain variance in the outcomes, but to identify individual factors that are potentially related to

	the various movement behaviors, and could be used to inform interventions. We are have now reported variances and ICC (see Table 1) as requested.
5. The presentation of the result in Table 3 is confusing. First, a typical journal table format does not use borders inside the table. Second, in the main text, the authors define 3 categories of school terms: winter, summer and spring. The table shows that the spring term is chosen as the reference category. Why is “school term” variable included as a different variable? The authors should specify the reference category for each categorical variable and present all non-reference categories under this same categorical variable. Please work on the discussion to consistent with your research questions as well as the terms used throughout the paper	We have removed all borders from the table 1, 2 and 3. The variable school term is included as data was collected during the school terms and holidays also during three seasons. We have added the reference category under each variable in table 4(previously table 3). We have added three paragraphs on our discussion on page 14.

Table 1: Performance of unconditional and full models for each outcome SB, LPA, and MVPA (new table in methods section)

Models	SB	LPA	MVPA
Nested vs. non-nested Likelihood ratio test	246.02 (p<0.001)	345.72 (p<0.001)	393.15 (p<0.001)
Unconditional models (school as a random effect)			
ICC	0.001	0.01	0.01
AIC	13991.08	12820.21	11379.40
BIC	14006.24	12835.37	11394.56
Log-likelihood	-6992.54	-6407.10	-5686.70
Unconditional models (child as a random effect)			
ICC	0.33	0.39	0.45
AIC	13745.28	12482.51	10996.79
BIC	13760.44	12497.67	11011.95
Log-likelihood	-6869.64	-6238.26	-5495.40
Full model (child as a random effect)			
ICC	0.33	0.35	0.37
AIC	12807.72	12097.39	10950.04
BIC	12.883.53	12173.19	11025.85
Log-likelihood	-6388.86	-6033.70	-5460.02